# Training-free Color-Style Disentanglement for Constrained Text-to-Image Synthesis

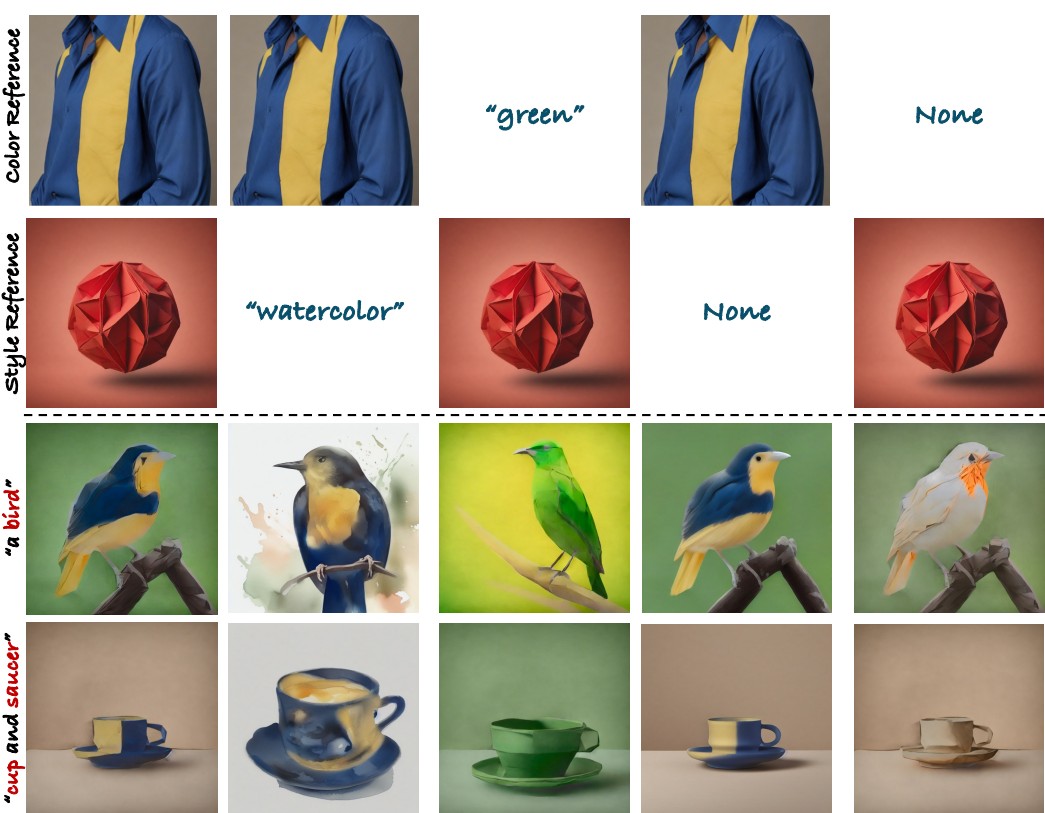

Figure 1. We propose the first training-free approach to allow disentangled conditioning of text-to-image diffusion models on color and style attributes from reference images.

## Abstract

We consider the problem of independently, in a disentangled fashion, controlling the outputs of text-to-image diffusion models with color and style attributes of a user-supplied reference image. We present the first training-free, test-time-only method to disentangle and condition text-to-image models on color and style attributes from reference image. To realize this, we propose two key innovations. Our first contribution is to transform the latent codes at inference time using feature transformations that make the covariance matrix of current generation follow that of the reference image, helping meaningfully transfer color. Next, we observe that there exists a natural disentanglement between color and style in the LAB image space, which we exploit to transform the self-attention feature maps of the image being generated with respect to those of the reference computed from its L channel. Both these operations happen purely at test time and can be done independently or merged. This results in a flexible method where color and style information

*can come from the same reference image or two different sources, and a new generation can seamlessly fuse them in either scenario (see Figure 1).*

## 1. Introduction

We consider the problem of conditioning the text-to-image class of diffusion models [12, 25, 30, 32] on color and style attributes extracted from a user-provided reference image. In particular, we want to independently control outputs of text-to-image models with either or both of these attributes, necessitating disentangled color and style conditioning. Furthermore, we seek to do so in a completely training-free and test-time-only fashion. This is practically an important problem since (a) such disentangled control means color and style information can now come from two different sources, and a new generation conditioned on them can be fused to produce an image with color from the first source and style from the second source, and (b) a test-time and training-free solution means one does not have to keep training models each time reference images change.

While there has been much work in customizing text-to-image generation [1, 8, 17, 26, 35, 47] with reference images, most of these techniques lack explicit control over which attributes from the reference are to be reflected in the synthesized images. Further, while there have been some attempts at training-free customization approaches, they all focus on a specific aspect, e.g., appearance transfer [2] or style transfer [11] or ensuring subject consistency [34]. None of these training-free methods are able to achieve disentangled attribute transfer that we seek to achieve in our work. Next, training-based methods such as MATTE [1] proposed a way to allow attribute-conditioned image synthesis but it needed (a) optimizing textual tokens that may take hours depending on compute and reference image, and (b) a separate custom loss function to achieve disentanglement between color and style. Some other training-based methods such as ProSpect [47], while doing multi-attribute conditioning, are also not able to disentangle color and style despite training tokens. Consequently, we ask, and answer affirmatively, two key questions- (a) can we achieve test-time-only conditioning of text-to-image models with color and style attributes from reference images? and (b) can we do (a) with disentangled control of color and style?

We begin with a brief discussion on why recent training-free methods such as [2] do not achieve disentangled attribute transfer. This method proposed to capture the customized concept by transfering keys and values computed from the reference image. Given observations from prior work [47] the color attribute is captured during the initial denoising stages and style in the later steps, a natural way to repurpose [2] for our task is to restrict these key-value operations to specific timesteps depending on the attribute

we seek to transfer. We show some results with this approach in Figure 2. As can be noted from these results, the attribute transfers are far from desirable. This is because color transfer using the key-value operations of [2] is limited by the quality of semantic correspondences between the reference image and the current generation. On the other hand, transferring style by simply limiting key-value copy to the last denoising timesteps is insufficient since by then features would have sufficiently entangled color and style information. Consequently, it is critical to disentangle these attributes in the feature space in a principled fashion to be able to achieve multi-attribute transfer at test time.

To address the aforementioned issues, we propose the first training-free method to enable disentangled control over color and style attributes from a reference image when generating new images. To extract and transfer the color attribute from a reference image, we propose to modify the latent codes during denoising using a novel correspondence-aware recoloring transformation. Our key intuition is there naturally exist color clusters in the reference image, and ensuring regions with the dominant color populations from reference correspond to regions in the image being generated will lead to meaningful color transfer. Given a color clustering of the reference image, we realize this by picking a certain denoising timestep, decoding the latent code, clustering the image, establishing a correspondences between the two sets of clusters, and performing a correspondence-aware whitening and recoloring feature transformation on the latent codes. At the end of the denoising process, the final latent code when decoded will give a new image with colors from the reference image. For instance, see first row/first column in Figure 1 where the bird follows the colors of the shirt. Next, to transfer style and disentangle it from the color attribute, we propose two innovations. First, we observe that there exists an inherent disentanglement between these two attributes in the LAB color space where the L channel contains content and style information and the A/B channels the color information [16]. See Fig 3 where we also show this qualitatively. In each row, we take the L channel from the image shown in the first column (after converting the RGB image shown into LAB), the A/B channels from the second column, and merge them, giving the result in the third column. In the first row, we see the style from the first column and the colors (blue) from the second column get captured in the resulting bird image. Similarly, in the third row, despite the second column having certain textured patterns, only the color (yellow) gets transferred to the resulting image. Next, noting that style mostly gets captured during the later denoising steps, we propose a time-step-constrained feature manipulation strategy. We do this by first generating an image with the baseline model and the desired prompt, and store the A/B channels. We then transfer style using the L channel from the reference image

Figure 2. Attribute entanglement in prior appearance transfer works [2].

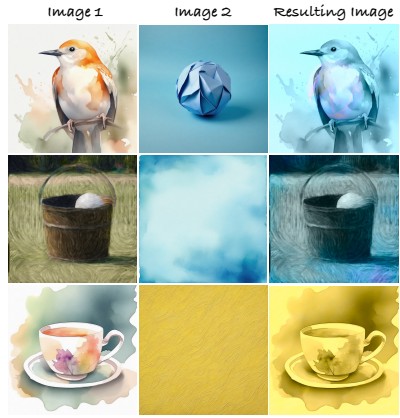

Figure 3. Attribute disentanglement in LAB space.

by aligning the self-attention key-value feature maps of the reference with those of the current generation, copy the A/B channels from the above operation and obtain the final result. See some results with our method in Figure 1 where in each case, our method is able to respect both color and style references in the final outputs. For instance, the first column has a "a bird" image following the blue/yellow colors from the color reference and origami style, the second column has the bird in blue/yellow colors and watercolor style, and so on.

To summarize, our key contributions in this work are:

- We present the first training-free method to disentangle and control text-to-image diffusion models on color and style attributes from a reference image.
- We propose a new time-step-constrained latent code recoloring transformation that aligns the covariance matrices of a text-to-image model output with that a reference image, helping transfer reference colors to outputs of text-to-image models.
- We notice that the L channel in the LAB space has an inherent separation between style and color and propose a new time-step-constrained self-attention key and value feature manipulation algorithm to transfer style from a reference image.

## 2. Related Work

With the wide adaptation of diffusion models for text-to-image synthesis [21, 24, 25, 27], much recent effort has been expended in controlling the outputs of these models. These efforts largely focus on learning adapters [20, 31, 44, 46] given baseline text-to-image models, finetuning parameters of the base model[17, 26, 33, 40], learning new tokens in the vocabulary of these models [1, 8, 47, 48], introducing dedicated personalization encoders [9, 18, 29, 39], utilising LoRA [15] to encapsulate target information [10, 19, 28, 38, 43, 49], and inpainting-based approaches [4, 42]. With the exception of MATTE [1], none of these methods are able to achieve disentangled transfer between color and style of a reference image. However, MATTE [1] needs custom loss functions and hours of training per reference image, which is practically infeasible.

On the other hand, there is a new line of recent work that involves training-free approaches to customization [2, 4, 5, 11, 34, 36, 37, 41, 45]. However, these methods focus on one specific aspect (e.g., appearance transfer, style transfer, or subject consistency) and are not able to provide independent disentangled control over color and style attributes. Similarly, the non-diffusion based conventional style transfer methods AesPA-Net[13], StyTR2[6], and ArtFlow[3] target disentanglement of content and style, while style and color are not treated independently/separately in these works. We address these gaps in both training-based and training-free methods by proposing the first training-free method to provide disentangled control for text-to-image models over color and style attributes from reference images. Moreover, our proposed approach is not specific to any dataset, and can seamlessly adapt to various styles/content due to the base model's capability.

## 3. Approach

We start with a brief review of latent diffusion models (LDMs). LDMs comprise an encoder-decoder pair and a

separately trained denoising diffusion probabilistic model (DDPM). Leveraging an encoder $\mathbf{E}$, LDMs translate an image $\mathbf{I}$ into a latent code $\mathbf{z}$, perform iterative denoising, and subsequently convert the predicted latent codes back to the pixel space via the decoder $\mathbf{D}$. The training objective of the DDPM $\epsilon_\theta$ is the following: $\mathbb{E}_{\mathbf{z}\sim\mathbf{E}(\mathbf{I}),p,\epsilon\sim\mathcal{N}(0,1),t}[\||\epsilon - \epsilon_\theta^{(t)}(\mathbf{z}_t,\mathbf{L}(p))\||]$ where p denotes any external conditioning factor e.g., a text prompt, which is typically encoded using text encoder $\mathbf{L}$ (e.g., CLIP [22], T5 [23]). At any timestep $t$ of the denoising process, given the current latent code $z_t$, the goal is to produce $z_{t-1}$. The first step here is to predict the noise $\epsilon_\theta^{(t)}(\mathbf{z}_t,\mathbf{L}(p))$. Given $z_t$ and $\epsilon_\theta^{(t)}(\mathbf{z}_t,\mathbf{L}(p))$, deterministic DDIM [32] sampling gives $z_{t-1}$ as

$$z_{t-1} = \sqrt{\alpha_{t-1}}z_0 + \hat{x}_t \tag{1}$$

where $z_0$ (denoised prediction) is predicted as $z_0 = \frac{z_t-\sqrt{1-\bar{\alpha}_t}\epsilon_\theta^{(t)}}{\sqrt{\bar{\alpha}_t}}$, and $\hat{x}_t$ (direction pointing to $x_t$) is computed as $\hat{x}_t = \sqrt{1 - \alpha_{t-1} - \sigma_t^2}\epsilon_\theta^{(t)}$.

## 3.1. Disentangled Color and Style Conditioning

As discussed in Section 1, we seek to provide text-to-image models with disentangled control over `color` and `style` attributes extracted from user-supplied reference images. To do this, we propose a new training-free algorithm that facilitates any of color-only, style-only, or both color-style transfer from reference images. We achieve this with a two-branch architecture (see Figure 4), one each for color and style. As we discuss later, outputs from these branches can be used independently (for single attribute transfer) or can be merged seamlessly. This merging can happen with color and style from the same source (one reference image) or color from one image and style from another image (see Fig 1 again where we show both single-source and two-source results).

Our closest training-free baselines [2, 34] inject key and value feature maps from self-attention blocks of the U-Net from the reference image. However, this only helps transfer the overall appearance/identity and cannot control color and style. To fix this gap, we propose two ideas. First, given color information from a reference image, we propose to apply recoloring transformations on the intermediate latent codes which when decoded can give an image following reference colors. Next, we notice that (a) style information gets captured only during the later parts of the denoising process and (b) style is captured in the L channel when an image is converted from the RGB to the LAB space (see Section 1 and Fig 3 again). We exploit these two observations to propose a timestep-constrained key and value feature manipulation strategy in the L space where features during early denoising steps are retained as is from the baseline generation and those of later steps are carried

over from the reference image. We next explain the details of these ideas.

**Color conditioning.** Our proposed method is visually summarized in the `color` branch in Figure 4. We first do a DDIM inversion step on the reference image to obtain the corresponding latent $z_t^{\text{ref}}$. Once the denoising process begins given a user-specified text prompt, given a latent $z_t$ at timestep $t$, the DDIM sampling will compute the noise prediction $\epsilon_\theta^{(t)}$, followed by computing the $z_0$ for both the reference image and the new generation. We then decode the latent code $z_0$ using the decoder $\mathcal{D}(.)$ (see Figure 5 for an example where we visualize these for several intermediate denoising timesteps). Given a timestep t, we first perform a K-Means clustering operation on both the decoded image $I_0^{(t)_{gen}}$ and the reference image to obtain sets of $K$ color clusters $\mathcal{C}_{\text{gen}}$ and $\mathcal{C}_{\text{ref}}$ respectively. Note that one can mask the decoded latent with cross-attention maps to restrict the object of interest in both the new generation as well as the reference image. Given the cluster sets, we next establish correspondences between them based on their proportion, giving a set of masks $\mathcal{M}_{ref}$ and $\mathcal{M}_{gen}$ for each set. The idea here is that a color cluster with the largest membership in the reference image indicates the dominant color that we seek to transfer to the current generation. Given the masks above, we achieve this by applying a mask-aware recoloring transformation (RT) on the latent code $z_0^{(t)_{gen}}$ as:

$$z_0^{(t)_{gen}} = \sum_{\substack{m_{gen}^i\in\mathcal{M}_{gen}\\m_{ref}^i\in\mathcal{M}_{ref}\\1\le i\le k}} \Big\{ \big(1 - m_{gen}^i\big)\,z_0^{(t)_{gen}} +$$
$$m_{gen}^i\Big[\text{RT}\Big(m_{gen}^i z_0^{(t)_{gen}}, m_{ref}^i z_0^{(t)_{ref}}\Big)\Big] \tag{2}$$

Here, we iterate over all the $K$ clusters and apply the recoloring transform separately to regions determined by masks corresponding to each cluster. In each iteration $i$, we use the corresponding mask $m_{gen}^i$ to constrain the region where color transfer happens, and similarly $m_{ref}^i$ determines the reference pixels from where the colors are picked. This way, we ensure that in any iteration $i$, pixels outside the region of interest (determined by the $m_{gen}^i$) remain untouched. The recoloring transformation [14] itself is a two-step process. We first whiten the latent codes to get its covariance matrix to be identity. We then apply a transformation to match the covariance matrix of the latent codes to match that of the reference image ($z_0^{(t)_{ref}}$). To ensure this operation strictly transfers color only and not style, we use observations from prior work [1, 47] that note color is captured during the early parts of the denoising process. Consequently, we restrict Eq 2 to only a subset of the initial denoising timesteps $t_{start}^c > t > t_{end}^c$.

The updated $z_0^{(t)_{gen}}$ obtained in Equation 2 is then used

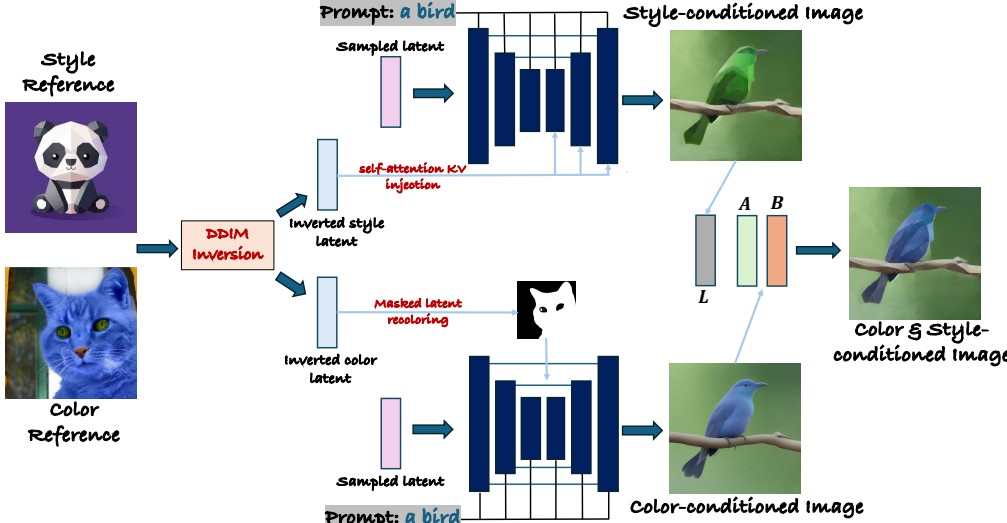

Figure 4. A visual illustration of the proposed method. (a) Obtain inverted latents for the style and color reference images, (b) Sample a new latent given a prompt (e.g., a bird above) and begin the denoising process for generating new images, (c) Perform *self attention KV injection* for style transfer, and/or *masked latent recoloring* for color transfer during this ongoing denoising process, (d) Utilise the intermediates obtained from style/color reference image reconstruction via style/color inverted latents respectively in step (c)

along with the predicted noise $\epsilon_\theta^t$ to compute $z_{t-1}^{gen}$ (using Equation 1) which goes as input to the next denoising step, eventually generating an image which follows the color from the reference image. We show an example demonstrating the progression of decoded latents $I_0^{(t)gen}$ across denoising timesteps in Figure 6. Here given the prompt a bird, one can observe that the model very initially starts forming some colors (green here). We transform the intermediate latents to manipulate these colors using the steps described above and obtain a blue bird following the blue cat in the color reference image shown in the figure.

**Style conditioning.** Given that high-frequency details like style and texture shows up in the later denoising timesteps [1, 47], we begin by injecting key and value feature maps from the reference image to the current generation for only the last few ($t > t_{start}^s$) denoising timesteps. However, an issue with this approach is the feature maps in the later timesteps will have color information as well (since color would have been captured in the beginning), leading to entanglement of color and style. See the "Resulting image (style repurposed)" column in Fig 2 for results with this approach- clearly both style and colors are getting transferred in this case, e.g., bird has both the watercolor style and the light pink colors from the "reference" image. To be able to disentangle style from color and allow independent control of the text-to-image diffusion model, our key insight is that there exists an inherent separation between style and color in the LAB space. The L channel captures the content and style, whereas the AB channels have color information (recall our discussion of Figure 3 in Section 1).

Since diffusion models are generally trained to operate in the RGB space, we take the grayscale version of the reference as an approximation to the L channel for all the next steps below (see Fig 4 again for a visual summary). We begin by DDIM inverting the reference to get the latent $z_t^{ref}$. Given any user-specified text prompt (e.g., a bird), for each denoising timestep $t < t_{start}^s$, we denoise the input latent codes as in a baseline text-to-image model but once we hit $t > t_{start}^s$, we start injecting the self-attention key $K$ and value $V$ feature maps from the reference reconstruction after converting it to grayscale and DDIM inverting it as noted above. Formally, this modified self-attention feature map computation at any denoising timestep $t$ and layer $l$ of the U-Net can be expressed as:

$$\hat{f}_t^l = 1_{0<=t<t_{start}^s}\text{softmax}\left(\frac{Q_{gen}^l K_{gen}^l{}^T}{\sqrt{d_k}}V_{gen}^l\right) +$$

$$1_{t>t_{start}^s}\text{softmax}\left(\frac{Q_{gen}^l K_{ref}^l{}^T}{\sqrt{d_k}}V_{ref}^l\right) \quad (3)$$

where 1 is an indicator, and $Q_{gen}^l/K_{gen}^l/V_{gen}^l$ and $Q_{ref}^l/K_{ref}^l/V_{ref}^l$ denote $l^{th}$ U-Net layer self-attention queries, keys, and values for the generation and reference respectively. We then take the final latent code, decode it to get an image, convert it to the LAB space, retain the L channel and get the AB channels from the corresponding color branch of Figure 4.

Note that $t_{end}^c$ ($= T/5$) is strictly less than $t > t_{start}^s$ ($= 4T/5$) throughout (i.e. the timestep intervals for which

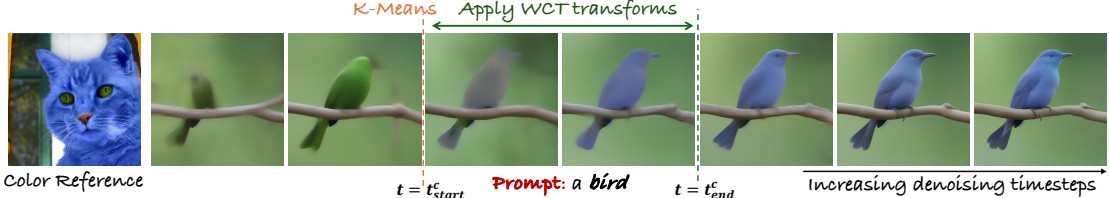

Figure 5. Intermediate decoded latents demonstrate the progression of color and fine-grained style information across denoising timesteps.

Figure 6. Progression of recolorised decoded latents across denoising timesteps.

color and style conditioning is applied have no overlap), where $T$ denotes the total number denoising steps.

## 4. Results

**Qualitative Evaluation.** In addition to the results in Figure 1, we show more results with our method in Figure 7 (color and style reference in the first two columns and our results in columns three-four) to demonstrate disentangled transfer of color and style attributes from reference images. We show different combinations of control over color and style attributes. In the first row, we are able to generate images following the content from the prompt (dog, vase) while following the style and color from the provided reference images. In the last two rows, our method generates images following style or color from the reference while imposing no control over the other attribute (observe the straight and sharp edges in dog's outline in the last row).

We next compare our method with various baselines including training-free style transfer (Cross-Image [2], StyleAlign [11], FreeDoM [45]) in Figure 8, conventional diffusion-free style transfer baselines (AesPA-Net[13], StyTR2[6], and ArtFlow[3] in Figure 10), training-free color transfer (Cross-Image [2], FreeDoM [45]) in Figure 11, and training-based techniques such as MATTE [1] and ProSpect [47] in Figure 12.

We begin by discussing disentangled style transfer results in Figure 8. One can note that our method is able to generate images (see last column) following the style from the reference image in a disentangled manner without affecting any other aspect of what the baseline generates (See Figure 9 for zoomed in section of images for the first/second rows of origami style transfer). On the other hand, as expected, the Cross-Image baseline transfers the full appearance of the reference image. One can also notice missing

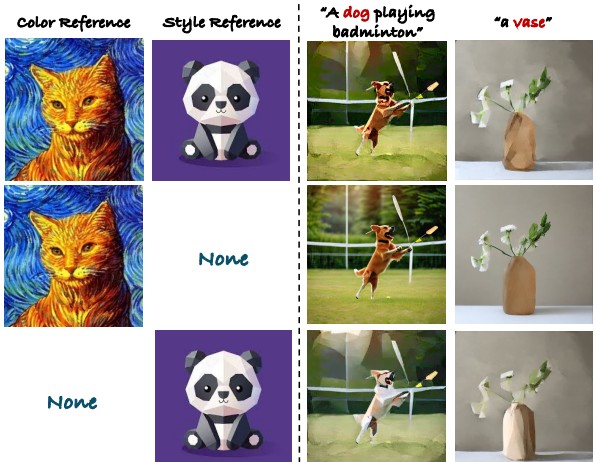

Figure 7. Disentangled color-style transfer from reference images.

regions in the bucket image in fourth row/fifth column due to lack of semantic correspondences. Similarly, the repurposed style version of cross-image baseline, while doing better, can not fully disentangle style from color as the features already have color information by the time style transfer happens during denoising. In StyleAlign, in addition to the style and color being entangled, the algorithm also transfers the structural/layout aspects of the reference image, thereby limiting the kind of control with style we seek over the final outputs (see fourth row/third column, where it tries generating bucket images following the layout of the cat from the reference image). Finally, FreeDoM also entangles color and style because its loss function does not account for any explicit disentanglement of these attributes.

We also compare with conventional style transfer methods in Figure 10. The baselines are able to preserve in-

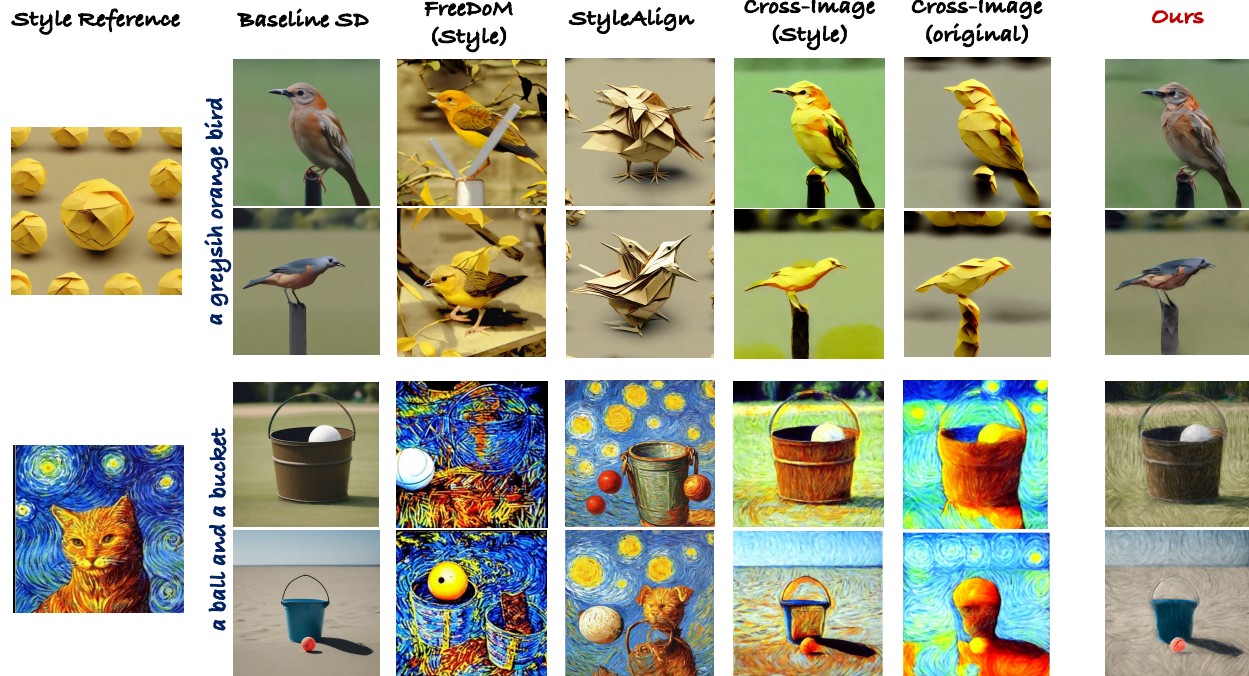

Figure 8. Comparing with state-of-the-art methods for disentangled style transfer. Please zoom in for viewing details.

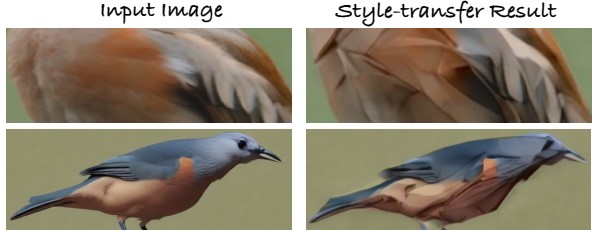

Figure 9. Style transfer (Origami).

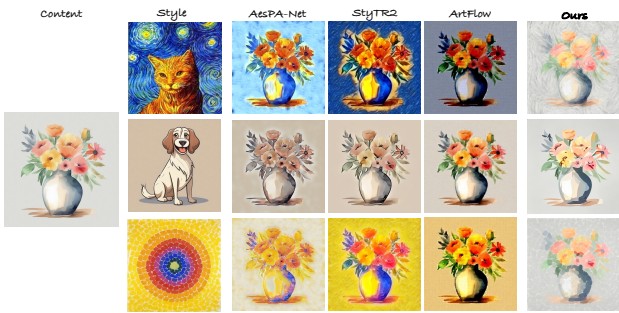

Figure 10. Comparison with conventional style-transfer baselines.

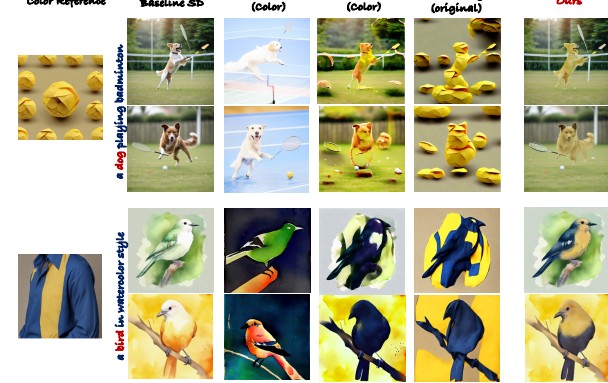

Figure 11. Comparing with state-of-the-art methods for disentangled color transfer.

put content, but the transfer of style (e.g. van-gogh patterns in first row) is limited. Since these methods target disentanglement of content and style, style and color are not treated independently/separately, leading to undesirable results (e.g. see the first row/second-third columns where the result flower vase has blue/orange colors from the style reference whereas our method in the last column is able to fix this issue). Finally, while these methods are trained on specific datasets, our method is training free and not specific to any dataset, and can seamlessly adapt to various styles/content due to the base model's capability.

We next discuss disentangled color transfer results in Figure 11. In all cases, our method is able to correctly transfer the color from the reference image whereas the cross-image (original) baseline transfers the full appearance from the reference image and can not control the color attribute independently (see third/fourth rows where the generated birds have a cloth-like appearance). Similarly, the repur-

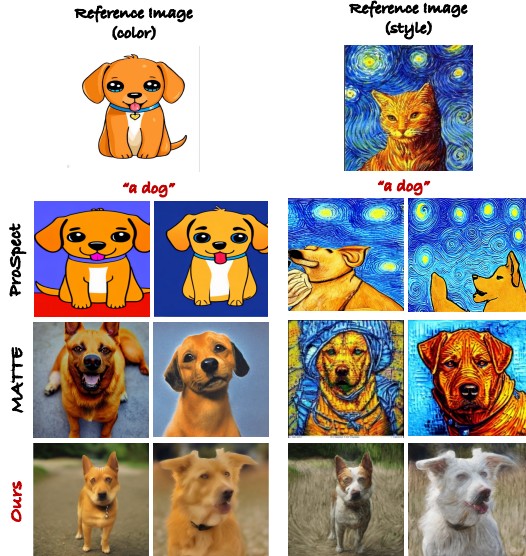

Figure 12. Comparing with recent state-of-the-art methods for multi-attribute constrained generation. Please zoom in for details.

| Method | color | style | disentanglement |
|---|---|---|---|
| ProSpect [47] | 0.26 | 0.26 | 0.19 |
| MATTE [1] | 0.27 | 0.27 | 0.26 |
| FreeDoM [45] | 0.20 | 0.24 | 0.19 |
| StyleAlign [11] | 0.26 | 0.25 | 0.20 |
| Cross-Image [2] | 0.25 | 0.26 | 0.20 |
| **Ours** | 0.28 | 0.26 | 0.26 |

Table 1. Comparison with baselines.

| Method | Preference |
|---|---|
| FreeDoM [45] | 4.9% |
| StyleAlign [11] | 7.1% |
| Cross-Image [2] | 18.7% |
| **Ours** | **69.3%** |

Table 2. User study results.

posed color version of cross-image baseline, while disentangling color and style, is unable to produce good results due to lack of correspondences. With FreeDoM, in some cases there is overfitting to the colors during optimization while disregarding the overall aesthetics, whereas in several other cases, the generated images disregards reference colors (the birds in last row/second column) due to incorrect optimization. Our method is able to control and transfer color attribute independently without affecting any other aspects of what the pretrained model would have generated.

Finally, in Figure 12, we compare to recent training-based methods. In ProSpect [47], one can see color and style are completely entangled (e.g., first column where both color and style are transferred). On the other hand, despite our method being completely training free, it performs at par with MATTE [1] which is a training-based approach (e.g., first column with our orange dogs). In the second column, whereas both MATTE and ProSpect entangle color and style, our method is able to generate van gogh-styled dogs without the bluish-orange colors.

**Quantitative Evaluation.** We next quantify improvements with our proposed method. We wish to evaluate how well these methods disentangle style and color (we follow the protocol from MATTE [1]), while also following the details specified as part of the text prompt. We keep either of the attributes (out of style/color) fixed from a reference image (we use the same set as in [1, 8, 35, 47]) and vary the other (we use the list of 7 style types, 13 object types and 11 color types from previous works [1, 35]). In each case, we synthesize a set of 64 images and compute the average CLIP image-text similarity. A higher score indicates better disentanglement since both attributes would then be

separately captured well in the output. To further evaluate the quality of transfer of each attribute, we also compute similarity scores between the ground truth color/style (color obtained using ColorThief [7]) and the generated images. As can be seen from Table 1, our method outperforms all training-free baselines and allows for independent control over style and color attributes. Further, when compared to the training-based MATTE [1], our method performs very competitively despite being training free. Finally, we conduct a user study with the generated images where we show survey respondents a textual prompt followed by color and style references, and ask them to select the images (among sets from four different methods) that best follow the provided constraints. From Table 2, our method's results are preferred by a majority of users, providing additional evidence for effectiveness of the proposed approach.

## 5. Summary

We considered the problem of disentangled color and style control of text-to-image models and noted none of the existing methods address this problem with a training-free approach. To this end, we proposed the first training-free, test-time-only solution with two key novelties: a timestep-constrained latent code recoloring transformation that aligned colors of generation outputs with reference colors and a timestep-constrained self-attention feature manipulation strategy in the L channel of the LAB space that aligned generation the style of generation outputs with that of the reference. This resulted in a flexible approach that can do color-only, style-only, or both color-style conditioning in a disentangled and indepedent fashion. Extensive qualitative and quantitative evaluations demonstrated the efficacy of our proposed method.

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
