# OpenReview forum: "Training-free Color-Style Disentanglement for Constrained Text-to-Image Synthesis"
_thecvf.com/CVPR/2025/Workshop/CVEU — CVPR 2025_

### Official Review · Reviewer_4B6G · 2025-03-14

**Rating:** 4
**Confidence:** 4

**Review:**

This paper proposes a pipeline to genertate images with style-reference and color-reference using diffusion models. For Style-reference, the authors use KV injection proposed in previous papers; for color-reference, they use clustering-based color transformation methods working on the intermediate timesteps. The experiments show that the pipeline is effective and is better than previous baselines.

Weaknesses:

The idea borrows idea from previous methods, and the generate images are not that fine grained. The task is new and a little bit restricted.

Strength:

The proposed framework can work in this specific task and show better performance than previous methods. Overally, I suggest this is a fit for this workshop.

---

### Official Review · Reviewer_TkgQ · 2025-03-14
**Reviews for Training-free Color-Style Disentanglement for Constrained Text-to-Image Synthesis**

**Rating:** 3
**Confidence:** 4

**Review:**

The paper presents an interesting observation that texture can be decoupled into color and style. From both the methodological perspective and the experimental results, the approach of treating an image as a combination of structure and texture, followed by further decomposition of texture into color and style, appears to effectively enable the generation of images that preserve structure while incorporating color and style from different reference images.

Furthermore, the proposed training-free method for simultaneous color and style transfer is promising. It eliminates the need for additional fine-tuning by leveraging image processing techniques and inherent model features to achieve color-style decoupling and generation.

However, the paper has several areas that require improvement:

1. Writing quality needs improvement: The overall writing quality is not very strong, particularly in critical sections such as the introduction, which appears unstructured and lacks careful organization. The introduction is overly lengthy, seemingly written without a clear logical flow. A more concise and well-structured introduction would improve readability.

2. Issues with the teaser figure: According to the framework (Fig. 4) and the methodology description, the structure of the image should remain unchanged even when integrating different color and style references. However, in Fig. 1, the second and third columns exhibit noticeable structural changes. This discrepancy needs to be clarified.

3. Weak decoupling of color and style: The impact of style on the image appears to be somewhat diminished compared to other style transfer methods. It may be worth exploring whether increasing the weight of the L-channel in the LAB color space, or employing other techniques, could enhance the influence of style in the generated images.

4. Low quantitative scores: Training-free methods often suffer from instability, and qualitative results could be cherry-picked. The paper would benefit from including some failure cases to highlight potential limitations and suggest future improvements.

5. Unclear explanation of LAB-based color-style decoupling: One major issue is the lack of an in-depth explanation of why LAB color space manipulation effectively achieves color-style decoupling. This omission makes it difficult for reviewers to fully grasp the motivation behind this approach and assess its reproducibility. A more thorough analysis in this regard would strengthen the paper.

Overall, the paper presents an intriguing direction but requires substantial refinement in both its writing and technical explanations to enhance clarity and impact.

---

### Official Review · Reviewer_L8Do · 2025-03-17
**interesting exploration on controlling color and style of generated images**

**Rating:** 4
**Confidence:** 4

**Review:**

Summary: The author introduces disentangled color and style conditioning for image generation. To achieve this goal, the author proposes two innovations: transform the latent codes to help meaningful color transformation; finding natural disentanglement between color and style in LAB image space.

Positives: 1. Results are good, as shown in figures 2 and 3. 2. Sufficient experiments

Neg: 1. Maybe the author can add the definition of LAB. 2. for example, in fig 4, the prompt (a bird), style and color refs as conditions for the image generation, how about comparing with directly using "a blue bird with xxx style" to generate image?

---

### Official Review · Reviewer_Yps9 · 2025-03-25
**Ambiguous problem definition; limited scalability beyond style and colors**

**Rating:** 2
**Confidence:** 3

**Review:**

Strengths:
1. The method is training-free and lightweight.
2. There are some empirical improvements upon prior methods on style-color disentanglement.

Weaknesses:
1. The paper does not provide a clear problem definition of style-color disentanglement. What's the definition of style? Would color be considered part of the style?
2. The method is tailored towards very specific controls, namely styles and colors, which restricts it from being applicable to further editing applications.
3. Style manipulation requires reducing the dimension of the conditioning style image into a 1-color channel, which may result in loss in the style details.

Minor issues:
1. Figures should be more clearly labeled. It's unable to tell which part of Figure 1 contains inputs and which part corresponds to outputs.

---

### Decision · Program_Chairs · 2025-03-25

**Decision:**

Accept

**Comment:**

The paper proposes a training-free method for disentangling color and style conditioning in text-to-image generation. Reviewers praised its lightweight nature, empirical improvements over baselines, and interesting exploration of style-color decomposition. However, they raised concerns about ambiguous problem definition, limited scalability beyond color and style, insufficient explanations of the LAB-based decomposition, and some structural inconsistencies in generated images.

Overall, the strengths outweigh the weaknesses, and the paper is accepted. Authors are encouraged to clearly define style vs. color, improve explanations on LAB-space manipulation, and clarify visual results in the camera-ready version.